# Socioeconomic disparities in attention deficit hyperactivity disorder (ADHD) in Sweden: An intersectional ecological niches analysis of individual heterogeneity and discriminatory accuracy (IEN-AIHDA)

Christoffer Hornborg[1,2,3]*, Rebecca Axrud[2], Raquel Pérez Vicente[2], Juan Merlo[2,4]

1 Department of Sociology and Work Science, University of Gothenburg, Göteborg, Sweden, 2 Unit for Social Epidemiology, Faculty of Medicine, Lund University, Malmö, Sweden, 3 Centre for Welfare, Social Innovation and Sustainability in Rural Areas, Campus Västervik, Västervik, Sweden, 4 Centre for Primary Health Care Research, Region Skåne, Malmö, Sweden

* Christoffer.hornborg@vastervik.se

## Abstract

We aimed (i) to gain a better understanding of the demographic and socioeconomical distribution of ADHD risk in Sweden; and (ii) to contribute to the critical discussion on medicalization, i.e., the tendency to define and treat behavioural and social problems as medical entities. For this purpose, we analysed the risk of suffering from ADHD in the whole Swedish population aged between 5 and 60 years, across 96 different strata defined by combining categories of gender, age, income, and country of birth. The stratified analysis evidenced considerable risk heterogeneity, with prevalence values ranging from 0.03% in high income immigrant women aged 50–59, to 6.18% in middle income immigrant boys aged 10–14. Our study questions the established idea that behavioural difficulties conceptualized as ADHD should be primarily perceived as a neurological abnormality. Rather, our findings suggest that there is a strong sociological component behind how some individuals become impaired and subject to medicalization.

## Introduction

In August 2022, The Moderate Party of Sweden proposed that children in vulnerable suburbs around the capital Stockholm should be offered rapid tests for attention deficit hyperactivity disorder (ADHD), as a preventive measure against exclusion and gang crime [1]. Without questioning the intentions of the proposal, there is an all the more important observation to make. Rather than addressing the social fabric surrounding children that grow up under adverse psychosocial circumstances, the proposal illustrated how the discourse of 'neuropsychiatry' has become more and more hegemonic in how to make sense of human behavior and social problems.

**Data Availability Statement:** All relevant data are within the paper and its Supporting Information files.

**Funding:** The authors received no specific funding for this work.

**Competing interests:** The authors have declared that no competing interests exist.

**Fig 1. Prevalence (%) of ADHD diagnosis in the population 5 to 60 year-old by sex and calender year.**

According to meta-analyses, the number of ADHD diagnoses has increased considerably all over the world [2]. Figs 1 and 2 summarizes data from the Swedish National Board of Health and Welfare [3], illustrating that from 2001 to 2018, the prevalence of ADHD diagnoses in open specialized care increased 36 times in boys and 112 times in girls between 5 and 19 years of age. This massive increase in the number of ADHD diagnoses during the last two decades is a notable epidemiological phenomenon [4]. Arguably, the threshold of who are considered to meet the criteria for a medical disorder has become lower, where parents, schools, and society in general seem to have become more inclined to use ADHD as an explanatory model when children display problematic or undesirable behaviour. In the dominant discourse on ADHD symptoms, the general emphasis is put on an underlying biological disturbance [5]. This is despite the fact that a tangible number of children who receive an ADHD diagnosis are subject to social disadvantage and stressful environments that may impact their development and well-being [6].

## The ontology of ADHD

Historically, the debate about ADHD has been rather polarized. In psychiatry and the natural sciences, ADHD has been conceptualized as a universally occurring medical condition that exists regardless of societal conditions and cultural settings [7]. Within this paradigm, researchers have sought to investigate neurobiological components associated with possible functional [8, 9], structural [10, 11], and neurotransmitter [12–14] alterations in various regions of the brain. Accordingly, ADHD has been treated as a *natural kind*, i.e., as a universal

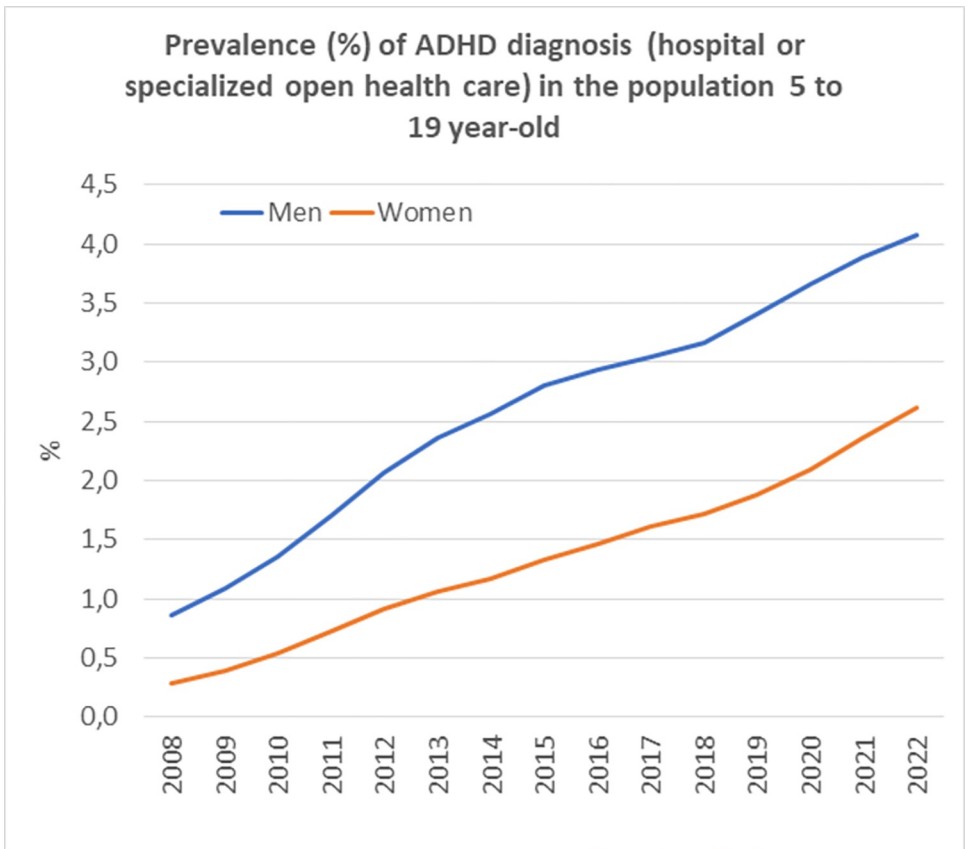

**Fig 2. Prevalence (%) of ADHD diagnosis in the population 5 to 19 year-old by sex and calender year.**

entity that exists independently of human systems of classification [15]. On the other hand, sociologists [16] and critical psychiatrists [17] have emphasised how societal and cultural processes have led to an expanded use of the diagnosis, and the scientific integrity of ADHD has also been questioned [18]. Others [19] have nuanced the tension by emphasizing how considering a diagnosis as a societal and cultural process, does not suggest that the diagnosis is not real in a specific context. Furthermore, the controversies around ADHD have been addressed as a problem of *reification*, i.e., the belief that if something is given a name, we have acquired an explanation [20]. Te Meerman and coworkers thus reminds us that "The descriptive classification Attention-Deficit/Hyperactivity Disorder (ADHD) is often mistaken for a disease entity that explains the causes of inattentive and hyperactive behaviors, rather than merely describing the existence of such behaviors". Accordingly, there is a risk that not only scientists, but also laymen and politicians tend to regard ADHD as an entity, rather than a mere name for observed behaviors. Such a point of departure does not diminish the suffering and seriousness of the symptoms, but when it comes to scientific understanding, problems of reification can illustrate the fallacies that systematically run through psychiatric thinking. As an example, in an analysis of the increased prescription of pharmacological treatment of ADHD, The Swedish National Board of Health and Welfare [21] stated that the proportion of people receiving a diagnosis eventually will be reflected in how many that *actually have* ADHD in the population, and when that happens, the curve for new users of medications will flatten out. The fallacy in this sort of thinking is that the diagnosis is conceptualized as a thing one simply has or not. There is a risk that an excessively narrow perspective on epidemiology will end up in a

discussion about whether ADHD is over- or underdiagnosed. Such a focus assumes that there exists an objective prevalence, but since most traits and behaviours, including those applied to define ADHD, are continuously distributed within the population, the prevalence is ultimately dependent on social consensus regarding the boundaries of what fits into the diagnosis and what does not. In other words, there is no 'natural' prevalence to be found, regardless of scientific rigour. ADHD as a category is so heterogenous, multi-factorial, and even ambiguous [22], that it could implicate an all too broad spectrum of idiopathic symptoms of suffering, depending on the intentions and bias of its advocates.

According to meta-analyses [23, 24], there has been no evidence to suggest an increase in the number of individuals who meet criteria for ADHD in the past three decades. Within the general Swedish population, there are no indications that 'ADHD traits' have increased, despite the obvious rise in the number of diagnoses. This discrepancy of course raises questions. Has society become better in *discovering* more cases of a valid disease entity, among individuals who have previously been undiagnosed? Or has it *changed* the view on what should be considered a disorder [25, 26]? Psychiatric diagnoses are flexible objects that arise, evolve, and even disappear, in line with historical processes [27, 28]. Something that started out as a label for hyperactive children, nowadays serve as a hypothetical cause for a tremendously broad range of behavioral and cognitive features, in both children and adults: difficulties with concentration, aggression, carelessness, burn-out syndrome, addiction, apathy, impulsivity, antisocial behavior, and so on–basically a majority of psychological attributes associated with impaired functioning. To treat ADHD as a more or less biological disease entity *causing* an impairment, can be a logical fallacy of reification, prohibiting a more nuanced understanding of human functioning. Therefore, it is important to map the contextual factors that are associated with individuals that become diagnosed. This is where sociological and philosophical perspectives have the potential to enrich the understanding of psychiatric epidemics such as ADHD [16, 27, 29–31].

## Ecological niches as a conceptual framework

Rather than seeking to establish the objective prevalence of ADHD, Hornborg and Merlo [25] have advocated that the focus should be to identify the way diagnoses and prevalence are moulded by how societal, economic, and ideological currents change the assumptions, conceptual apparatus and explanatory models of clinicians and researchers. What type of social and cultural forces may have increased the prevalence and in which settings do ADHD occur? One way of understanding social and psychiatric epidemiology on the basis of these issues is by applying the philosophy of Ian Hacking [27]. Instead of perceiving controversial diagnoses as either medically valid or merely socially constructed, Hacking sought to understand the contextual factors that allowed the diagnosis to arise and thrive over time and space. To do this he used the concept of ecological niche, a metaphor derived from biology. Just as biological organisms only will survive in a niche with the right conditions, the same applies to certain types of diseases. For example, the diagnosis of dyslexia would not flourish in a society without a written language. It is possible to argue that the symptoms of ADHD have always existed, but the idea of an ecological niche helps to illuminate in which conditions they become classified as a medical diagnosis. Hacking's metaphor is epidemiologically relevant since it acknowledges both a disorder (analogous to the organism) and its context (defined by the factors that forms the ecological niche in which the organism thrives), and it can be particularly useful in social epidemiology. In drastic epidemiological changes, it is of great importance for research to study the contextual forces that underly the act of diagnosing. This can be done by investigating vehicles on a macro level, such as the influence of the pharmaceutical industry, an

increased biological focus within psychiatry, transitions from ICD to DSM and the role of the Internet [32], i.e., factors that affect the entire way research is conceptualized, as well as the general perception of laymen and media. But it can also be done by examining how different micro niches, domains within society, is affected differently by the changing perceptions of what constitutes a disorder, acknowledging how the act of turning human behaviour into disease always takes place within a particular context. If the national ADHD prevalence has a considerable heterogeneity in the distribution of diagnoses, this could be framed as an existence of smaller ecological niches in which demographical and socioeconomic factors condition the setting where diagnoses occur. This approach is analogous to the traditional study of inequalities in health where socioeconomic differences are investigated with an intersectional approach [33].

## Intersectional analysis of individual heterogeneity and discriminatory accuracy (AIHDA)

Intersectionality was originally a theory that mainly applied to qualitative methodology. However, an intersectional quantitative approach is being increasingly utilized in epidemiology and public health [33–37] as it provides an ideal framework for understanding the demographic and socioeconomic heterogeneity in diagnoses existing within a community. The Oxford English Dictionary 2015 defines intersectionality as "*The interconnected nature of social categorizations such as race, class, and gender, regarded as creating overlapping and interdependent systems of discrimination or disadvantage*". Thereby, the demographic, and socioeconomic dimensions that condition health and disease risk need to be considered as interlocked rather than as unidimensional. By using intersectionality, numerous intersectional groups (i.e., strata) can be defined by a *combination* of several demographic and socioeconomic dimensions (e.g., age, gender, income, and country of birth). In this way, the intersectional perspective provides an improved mapping of disadvantage that better illustrates the heterogeneous socioeconomic distribution of health in the community. Such intersectional strata could be conceptualized as micro niches that facilitate or hinder the construction of ADHD diagnoses by providing different thresholds for dichotomizing traits and behaviours. For instance, many of the questions included in self-assessment forms for ADHD such as ASRS-v1.1 [38] can be interpreted differently in different intersectional contexts. Also, schoolteachers and legal guardians from different niches may have different expectations that condition the evaluation of the symptomatology [39–41]. The influence of intersectional niches could be perceived as an addition of influences from separate social dimensions or as an interaction where the propensity of receiving an ADHD diagnosis is larger or lower than the sum of the influences of separate social dimensions.

The present study develops an intersectional perspective into an intersectional ecological niche (IEN) framework. The IEN framework benefits from intersectionality by allowing an improved use of interlocked socioeconomic dimensions, but simultaneously allows the considerations of whatever other relevant ecological factors that would not traditionally be included in a formal intersectional analysis. Thereby, the IE framework gains flexibility when it comes to investigating overlapping and interdependent systems of discrimination or disadvantage. Utilizing the concept of ecological niches is a way of counteracting essentialist narratives and developing a more constructionist framework on how disorders such as ADHD flourish due to contextual factors. Instead of conceptualizing ADHD as a universally occurring entity, it is approached as something that emerges more or less in a given context. Accordingly, the IEN-AIHDA framework that we propose may be a way of addressing the emergence of mental illnesses *both* in societies and in individuals [cf. 42].

By applying a methodology denominated *multilevel analysis of individual heterogeneity and discriminatory accuracy* (MAIHDA) [36], the individual is considered as nested within intersectional ecological niches. Such a methodology can be applied by using both formal multilevel regression [43, 44] as well as traditional regression analyses [33]. MAIHDA provides a detailed mapping of the risk of ADHD diagnosis in the population and allows an evaluation to which degree micro niches are with accuracy able to discriminate individuals with ADHD from individuals without ADHD [36, 45]. Traditional methodologies assign the same average value (i.e., probability) to all the members in the group (i.e., intersectional niche) without considering the individual heterogeneity around the group average and the overlapping between categories [46]. If the discriminatory accuracy is low, the benefit of focusing on specific intersectional niches for understanding differences in ADHD diagnosis can be discussed.

## Aim

The aim of this study is twofold. Firstly, we want to obtain a better understanding of the demographic and socioeconomic distribution of ADHD diagnosis in the Swedish population. We hypothesised that different intersectional niches (i.e., contexts, strata) may promote or hinder the risk of getting an ADHD diagnose in Sweden. That is, the same individual would receive or not receive the diagnosis if he/she was exposed to a different niche. If this is true, differences between the intersectional niches would be high. Secondly, we want to contribute to the ongoing critical discussion on medicalization, i.e., the tendency to increasingly define and treat behavioural and social problems as medical entities. Our analysis of intersectional niches is therefore grounded in a nominalist point of departure, highlighting the importance of critically examining current discourses on disease categories such as ADHD.

- How is the prevalence of ADHD diagnoses distributed across intersectional niches defined by gender, age, country of birth, and income?

- What is the discriminatory accuracy of the intersectional niches for classifying individuals regarding the existence of an ADHD diagnosis?

- In what way can the findings contribute to the ongoing discussion about ADHD as a historically situated diagnostic trend of epidemic proportions?

## Methods

### Databases

We used a database composed of three linked registers covering the whole country of Sweden: The Register of the Total Population (TPR) [47], the Longitudinal Integration Database for Health Insurance and Labour Market Studies (LISA) administered by Statistics Sweden [48], and The National Patient Register (NPR) [49] administered by the National Board of Health and Welfare [50]. We performed a record linkage using the unique 12-digit social security number (SSN) assigned to each individual residing in Sweden. In order to pseudonymize the research database, Statistics Sweden in coordination with The National Board of Health and Welfare replaced the SSN with an arbitrary serial number before delivering files. The information is registered without formal individual consent. However, while the data is confidential, the Swedish authorities delivers pseudonymize information for research purposes after evaluation.

The TPR [47] registers data on all Swedish residents regarding socioeconomic information about gender, civil status, migration, country of birth and family relationship as well as general

demographics for those who stayed in Sweden for at least 12 months. The LISA database is administered by Statistics Sweden [48] and combines information on all Swedish residents above the age of 15 from several registers, including the labour market, social sectors, and educational sectors. It includes status on living conditions, demographics, and socioeconomic factors, like income, education, and employment. The NPR [50, 51] records more than 99% of inpatient and approximately 80% of specialized outpatient diagnoses from hospitals in Sweden. At the time of the study, all diagnoses were coded according to the International Classification of Diseases, 10th version (ICD-10). The NPR collects data from both private and public health care providers with an 85–95% positive predictive value on inpatient diagnosis [49]. Data from primary care is not included in the NPR.

## Study population

According to the TPR, 9.420.128 individuals resided in Sweden at the baseline date of December 31st, 2010 (Fig 3). When comparing data with the Death Register, we excluded 5.630 individuals that had passed away but with delayed registration, making the correct number of individuals 9.414.498. Based on the TPR, 444.596 individuals that had stayed in Sweden for less than 5 years before baseline were excluded, since recently arrived immigrants might have incomplete register information. Finally, we excluded 2.882.109 individuals older than 60 or younger than 5 years, leaving the database containing information on 6.093.423 individuals.

## Assessment of the outcome variable

The outcome variables used were presence or absence of ADHD diagnosis between 2005 and 2010. We defined ADHD diagnosis as any variant of International Classification of Diseases, 10th version (ICD-10) codes F90.0-F90.8.

## Assessment of demographic and socioeconomic variables

*Gender* was binary coded as either man or woman according to legal status. We categorized *age* at the baseline date into eight intervals: 5–9, 10–14, 15–19, 20–24, 25–29, 30–39, 40–49 and 50–59 years of age. It is known that an ADHD diagnosis is more common in boys than girls [52–57] and studies have also concluded that it is more common amongst 10–14-year-old children [58, 59]. Furthermore, we used information from Statistics Sweden to classify *country of birth* into a binary variable. We labelled individuals born in Sweden as 'natives' and individuals born outside of Sweden as 'immigrants'.

In order to achieve accurate measures of the individuals' income, the *income* variable was created using the individualized cumulative disposable income of the household. The disposable income is the income that remains after taxes has been accounted for. We computed a variable for the years 2000, 2005, and 2010, and divided the total disposable income of a family by the number of individuals in that family. The different consumption weights of adults and children of different ages were used according to a specific formula [60]. We sorted groups by 25th quantiles using the TPR in the years 2000, 2005, and 2010. Thereafter, were added the scores from the three years ranging from 1 to 25 and obtained values from three (lowest cumulative income) to 75 (highest cumulative income). Finally, we categorized the cumulative income into three groups (high, middle, and low) by tertiles. The 1002 individuals with missing values of income during 2000 or 2005 were assigned the tertile values of the year 2010. No individuals in our study population had missing income data in that particular year.

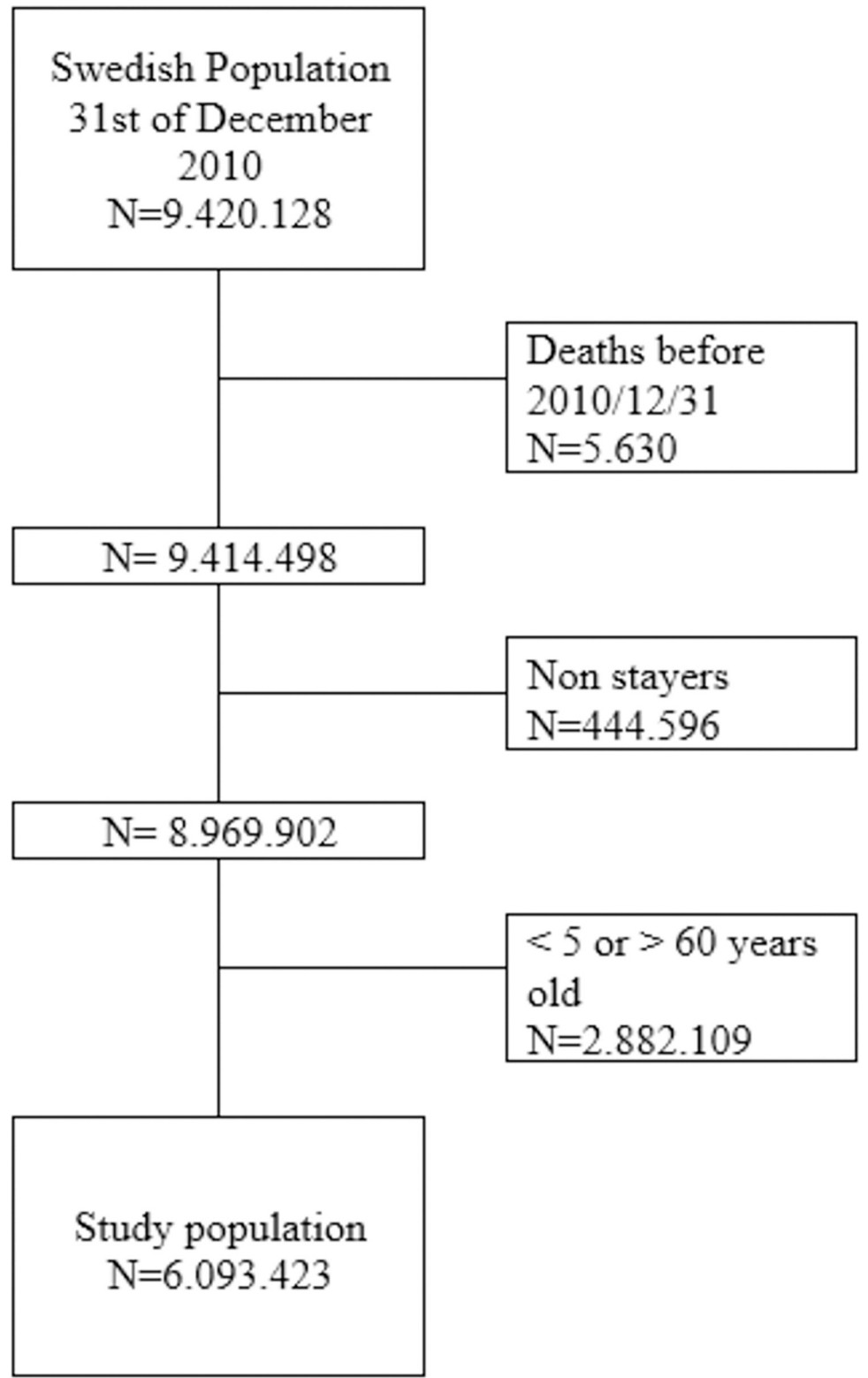

**Fig 3. Flow chart informing on inclusion and exclusion criteria, as well as number of individuals included in the study population.**

## Assessment of intersectional niches

As a way of including the intersectional niches, we created a multi-categorical variable consisting of 96 strata. The strata were created by combining the two categories of gender, the eight categories of age, the two categories of country of birth and the three categories of cumulative income. We used a reference stratum constructed of *10-14-year-old women born in Sweden with high income* for comparison. This choice was somewhat arbitrary but based on previous studies indicating the characteristics of the individuals with the lowest ADHD risk.

## Statistical analysis

The first step in our analysis was to map the prevalence (i.e., absolute risk) of ADHD across the intersectional niches by graphical stratified analyses. Thereafter, we modelled the binary ADHD variable as the dependent variable. We applied Cox proportional hazard regressions with a constant follow-up time equal to one. In this way, we formally attained prevalence ratios (PR) or relative risks [61].

We performed six different models. Model 1 included only age, model 2 only gender, model 3 only income, and model 4 only country of birth. In model 5 all four single variables were entered simultaneously. Finally, in model 6, we included the same information as in model 5 but with the multi-categorical variable with the stratum of *10–14-year-old native women with high income* as reference. In all models we calculated PR with 99% instead of 95% confidence intervals (CI) to counteract the multiple comparisons problem.

For each model, we quantified its discriminatory accuracy (DA) by means of the area under the receiver operator characteristics curve (AUC) [45]. The AUC is calculated by plotting the true positive fraction (i.e., sensitivity) against the false positive fraction (i.e., 1 –specificity) for different binary classification thresholds of the predicted probability of ADHD diagnosis. Thus, the AUC measures the accuracy of the information provided by the variables in the model used to discriminate individuals with ADHD from those without ADHD. The AUC obtains a value between 0.5 and 1, where 1 indicates perfect discrimination and 0.5 means that the studied variables have no discriminatory accuracy at all. Rather than only evaluating the differences between absolute risks of strata values, the DA also assessed the overlapping of the individual risk predictions (based on the intersectional niches) between individuals with and without ADHD. There is no fully established practical guideline for the interpretation of the size of the AUC as a measure of discriminatory accuracy when analysing IEN inequalities. However, based on the classification provided by Hosmer and Lemeshow [62] we defined the discriminatory accuracy as (i) 'absent or very low' (AUC = 0.5–0.6) corresponding with 'absent or very low' strata inequalities, (ii) 'poor' (AUC $>$0.6–$\leq$ 0.7), corresponding with 'small' strata inequalities, (iii) 'acceptable' (AUC $>$0.7– $\leq$ 0.8), corresponding with 'large' strata inequalities, (iv) 'excellent' (AUC $>$0.8–$\leq$ 0.9) or (v) 'outstanding' (AUC $>$ 0.9–1) corresponding with 'very large' strata inequalities.

We further calculated the gradual change in the AUC value (Δ-AUC) between the models. The Δ-AUC quantifies the improvement in the discriminatory accuracy obtained by a model, in relation to a reference model [63]. The categorical intersectional variable in model 6 allows us to capture possible interaction of effects between the variables that define the strata so if any interaction exists, the discriminatory accuracy of model 6 in comparison with model 5 will increase and the Δ-AUC be positive [33].

All statistical analyses were performed using IBM SPSS (Statistical Package for the Social Sciences) version 24 and Stata v15 (StataCorp, College Station, TX).

## Results

Out of all 6.093.423 individuals registered in our database, 54.181 (0.9%) were diagnosed with ADHD during the study period. Table 1 shows that the prevalence of ADHD diagnoses increases with age from 0.72% among 5–9-year-old children until the age of 15–19 when it reaches its maximum of 2.32%, and thereafter decreases with age reaching its lowest value of 0.2% among people 50–59 years old. Overall, ADHD is more frequent in men and in native Swedes than in women and immigrants. Also, there is a clear income gradient with the highest prevalence in the low-income group. Figs 4 and 5 state the prevalence in the different strata using two different approaches, in order to improve our understanding of the underlying heterogeneity in the distribution of ADHD diagnoses across different IE niches. The principal finding is that such heterogeneity is considerable. When observing the stratified analysis of the 96 groups, the risk of ADHD is *higher* in most immigrant strata than in native strata, especially in males. However, Table 2, which does not contain stratified information, indicates that immigrants have a *lower* ADHD risk than natives. This could illustrate the so-called Simpsons paradox [64], which describes that when combining strata, a trend can disappear, or show the opposite outcome in comparison to unidimensional analysis.

The tables and graphs demonstrating the strata or IE niches, present a similar pattern as in Table 1 concerning the prevalence of different age intervals as well as a higher prevalence in men than women. However, according to the strata seen in Figs 4 and 5, the income gradient appears to show a different pattern for immigrants compared to native Swedes. The immigrants with low income show the lowest prevalence amongst the income groups until the age of 24, thereafter the income gradients disappear, and the prevalence appears to be low in all three income groups. In contrast, the native Swedish groups present an income gradient across all age intervals and show a lower prevalence amongst individuals with high income. Observing the native Swedes, the absolute risk of having an ADHD diagnosis amongst young boys (10–19 years old) with a low income compared to the boys in the high-income groups is almost three times higher. The outcome is similarly much higher in immigrant boys when comparing them to native Swedish boys, in the high- and middle-income groups.

**Table 1. Number of individuals (N), number of individuals with Attention Deficit Hyperactivity Disorder (ADHD) (n) and prevalence of ADHD in Sweden between 2005 and 2010.**

|  |  | n | N | Prevalence (%) |
|---|---|---|---|---|
| **Sweden** |  | **54 181** | **6 093 423** | **0.89** |
| Age (years) | 5–9 | 3502 | 485 595 | 0.72 |
|  | 10–14 | 9742 | 460 644 | 2.11 |
|  | 15–19 | 13694 | 589 363 | 2.32 |
|  | 20–24 | 8279 | 581 506 | 1.42 |
|  | 25–29 | 4594 | 502 999 | 0.91 |
|  | 30–39 | 6804 | 1 102 084 | 0.62 |
|  | 40–49 | 5342 | 1 239 052 | 0.43 |
|  | 50–59 | 2224 | 1 132 180 | 0.20 |
| Gender | Female | 18556 | 2 991 959 | 0.62 |
|  | Male | 35625 | 3 101 464 | 1.15 |
| Country of birth | Immigrant | 3709 | 727 362 | 0.51 |
|  | Native | 50472 | 5 366 061 | 0.94 |
| Income | Low | 33299 | 2 101 824 | 1.58 |
|  | Middle | 16371 | 2 218 311 | 0.74 |
|  | High | 4511 | 1 773 288 | 0.25 |

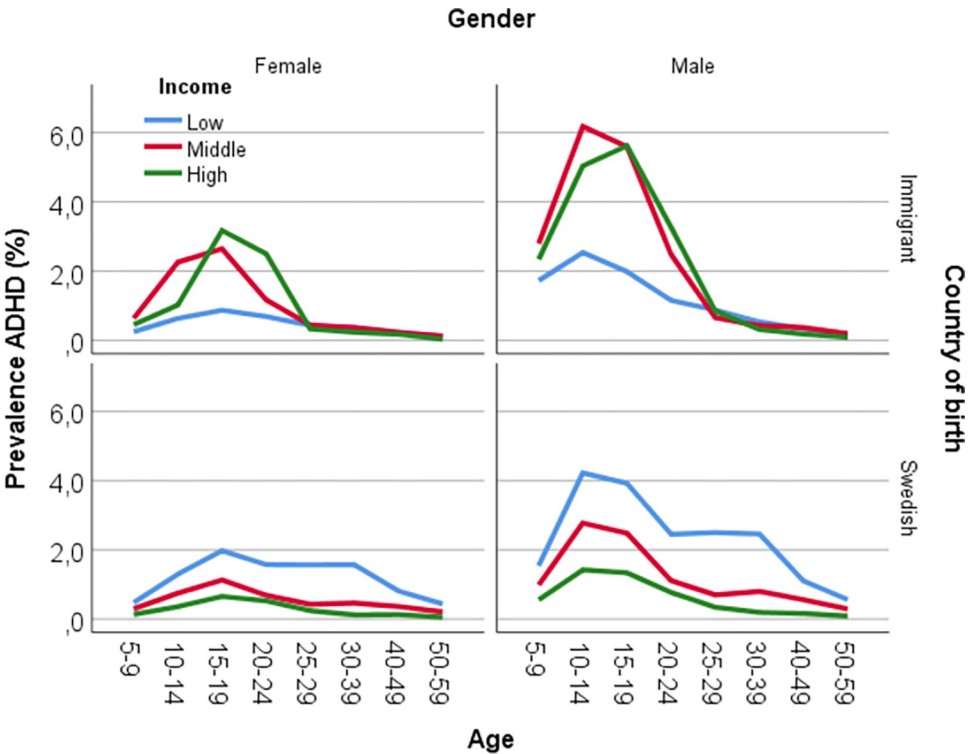

**Fig 4. Prevalence (absolute risk) of Attention Deficit Hyperactivity Disorder (ADHD) by gender, age, country of birth and low (L), middle (M) and high (H) income in the study cohort.**

Table 2 displays the PR indicating the association of ADHD and age (Model 1), gender (Model 2), income (Model 3) and country of birth (Model 4) in separate analyses. It also displays the variables stated above put together as one specific variable (Model 5), and as multi-categorical variable with 96 strata (Model 6). The results of model 6 are displayed in Table 3

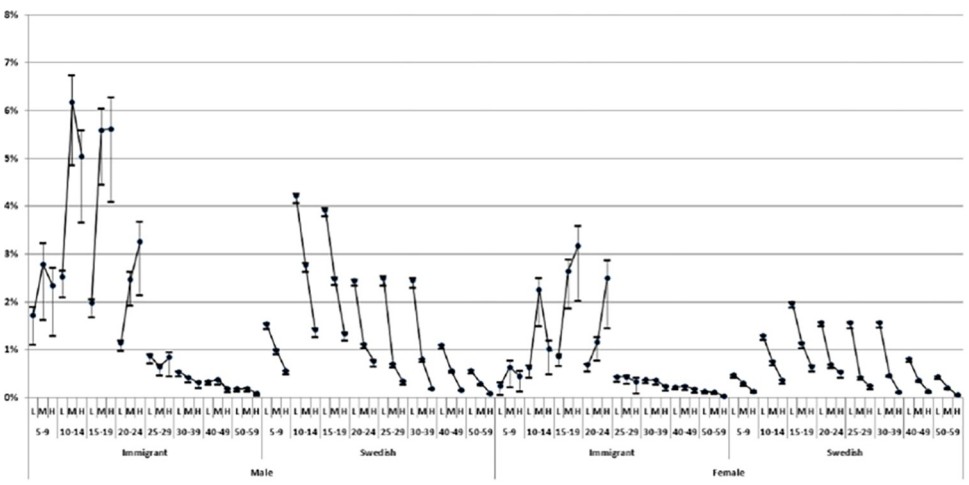

**Fig 5. Prevalence (absolute risk) of Attention Deficit Hyperactivity Disorder (ADHD), black circles across the 96 intersectional strata defined by gender, country of birth, age and low (L), middle (M) and high (H) income.** The lines crossing the circles are 99% confidence intervals. The lines between the strata show the income gradient across each age interval.

**Table 2. Prevalence ratios [a].**

| | | Model 1 | Model 2 | Model 3 | Model 4 | Model 5 | Model 6 |
|---|---|---|---|---|---|---|---|
| | N | (99.0%CI) | | | | | |
| Age (years) | 5–9 | 0.34 (0.32–0.36) | | | | 0.38 (0.36–0.40) | |
| | **10–14** | **Ref** | | | | **Ref** | |
| | 15–19 | 1.10 (1.06–1.14) | | | | 1.03 (1.00–1.07) | |
| | 20–24 | 0.67 (0.65–0.70) | | | | 0.65 (0.62–0.67) | |
| | 25–29 | 0.43 (0.41–0.45) | | | | 0.50 (0.48–0.53) | |
| | 30–39 | 0.29 (0.28–0.30) | | | | 0.44 (0.42–0.46) | |
| | 40–49 | 0.20 (0.20–0.21) | | | | 0.28 (0.27–0.29) | |
| | 50–59 | 0.09 (0.09–0.10) | | | | 0.14 (0.13–0.15) | |
| Gender | **Female** | | **Ref** | | | **Ref** | |
| | Male | | 1.85 (1.81–1.90) | | | 1.93 (1.89–1.98) | |
| Income | Low | | | 6.23 (5.98–6.49) | | 4.56 (4.37–4.76) | |
| | Middle | | | 2.90 (2.78–3.03) | | 2.40 (2.30–2.51) | |
| | **High** | | | **Ref** | | **Ref** | |
| Country of birth | **Swedish** | | | | **Ref** | **Ref** | |
| | Immigrant | | | | 0.54 (0.52–0.57) | 0.57 (0.54–0.59) | |
| | AUC | 0.76 (0.76–0.76) | 0.57 (0.57–0.58) | 0.67 (0.66–0.67) | 0.53 (0.52–0.53) | 0.76 (0.76–0.76) | 0.77 (0.77–0.77) |

[a]Prevalence Ratios (PR) and 99% confidence intervals (CI) indicating the association between age (Model 1), gender (Model 2), income (Model 3) and country of birth (Model 4) in separate analyses and in analyses including all those variables together as specific variables (Model 5) and as a multicategorical variable with 96 strata (Model 6). The table also informs on the area under the ROC curve and 99% confidence intervals (CI) obtained in models 1 to 6.

and show the prevalence or absolute risk for the 10 highest and 10 lowest strata out of all 96 that are seen in the supplementary information (S1 Table). These strata are compared to the reference *10-14-year-old women born in Sweden with high income*.

When comparing the 96 strata, the difference in absolute risk, AR, is striking. 10-14-year-old immigrant boys had an AR of 6.18, which is more than 200 times as high as that of 50-59-year-old immigrant women (AR = 0.03). As illustrated in Table 3, the risk was highest among middle- and high-income immigrant boys between the age of 10 and 19, followed by low-income native boys in the same age group. For full information, see S1 and S2 Tables.

The AUC-value, hence the DA, was calculated to 0.76 when all four variables were included in Model 5. Comparing this value with the models including only one variable at time showed no difference in age (Δ-AUC = 0.00; 0.76–0.76) but a difference compared to gender (Δ-AUC = 0.02), income (Δ-AUC = 0.09) and country of birth (Δ-AUC = 0.23). The DA for model 6 was calculated to be 0.77 which indicates a difference and a small interaction effect comparing it to model 5 (Δ-AUC = 0.01). Thus, the variables used in our study present mainly additive effects and only a small interactional effect.

## Discussion

Adopting an IEN approach, this study provides a detailed stratified analysis that offers a better understanding of the ADHD 'epidemic' in Sweden in 2010. It shows that behind the overall national prevalence there is considerable demographic and socioeconomic heterogeneity in the distribution of ADHD diagnoses in the Swedish population. As expected, the risk of receiving a diagnosis had a peak in younger age groups and was higher amongst males. However, the stratified analysis illustrates how this risk is strikingly higher in young immigrants than in natives, especially in males. It is noteworthy that, overall, natives had a higher absolute risk of

**Table 3. Number of individuals (N) and prevalence or Absolute Risk (AR) of Attention Deficit Hyperactivity Disorder (ADHD) in the 96 intercategorical strata as well as prevalence ratios (PR), obtained in model 6.** Only the 10 lowest and 10 highest PR are shown.

| | N | AR (99% CI) | PR (99% CI) |
|---|---|---|---|
| Female 50–59 High Immigrant | 25 934 | 0.03 (0.01–0.04) | 0.09 (0.03–0.22) |
| Female 50–59 High Swedish | 218 100 | 0.05 (0.04–0.05) | 0.15 (0.11–0.21) |
| Male 50–59 High Immigrant | 20 962 | 0.09 (0.04–0.09) | 0.24 (0.13–0.46) |
| Male 50–59 High Swedish | 235 261 | 0.09 (0.07–0.09) | 0.24 (0.18–0.32) |
| Female 50–59 Middle Immigrant | 29 176 | 0.12 (0.07–0.12) | 0.32 (0.20–0.53) |
| Female 30–39 High Swedish | 156 217 | 0.12 (0.10–0.12) | 0.33 (0.25–0.45) |
| Female 50–59 Low Immigrant | 38 058 | 0.13 (0.08–0.13) | 0.35 (0.23–0.54) |
| Female 40–49 High Swedish | 152 524 | 0.13 (0.11–0.13) | 0.36 (0.27–0.48) |
| Female 5–9 High Swedish | 57 851 | 0.13 (0.10–0.14) | 0.37 (0.26–0.54) |
| Male 40–49 High Swedish | 236 364 | 0.16 (0.14–0.16) | 0.44 (0.34–0.57) |
| Reference: Female 10–14 High Swedish | 37 267 | 0.36 (0.28–0.37) | Ref |
| Male 10–14 Middle Swedish | 81 227 | 2.77 (2.63–2.81) | 7.71 (6.13–9.70) |
| Male 5–9 Middle Immigrant | 1 002 | 2.79 (1.63–3.24) | 7.77 (4.55–13.27) |
| Female 15–19 High Immigrant | 1 229 | 3.17 (2.03–3.60) | 8.83 (5.52–14.10) |
| Male 20–24 High Immigrant | 1 318 | 3.26 (2.13–3.68) | 9.07 (5.78–14.25) |
| Male 15–19 Low Swedish | 154 354 | 3.92 (3.80–3.96) | 10.91 (8.71–13.66) |
| Male 10–14 Low Swedish | 105 259 | 4.22 (4.06–4.27) | 11.74 (9.37–14.72) |
| Male 10–14 High Immigrant | 1 449 | 5.04 (3.67–5.59) | 14.01 (9.63–20.38) |
| Male 15–19 Middle Immigrant | 2 397 | 5.59 (4.45–6.04) | 15.55 (11.35–21.30) |
| Male 15–19 High Immigrant | 1 282 | 5.62 (4.09–6.27) | 15.62 (10.72–22.76) |
| Male 10–14 Middle Immigrant | 1 959 | 6.18 (4.86–6.73) | 17.18 (12.44–23.73) |

ADHD than immigrants, yet when performing the stratified analysis, many of the immigrant strata had a higher prevalence than the native strata. Also, while there is a typical income gradient in native Swedes (with a higher ADHD risk in low-income individuals), a different pattern is observed among immigrants, where the low-income group has lower risk than middle- and high-income groups.

A map of the distribution of ADHD diagnoses across numerous IE strata is useful for identifying contexts with a potential higher risk. However, a disadvantage is that this detailed information may promote the stigmatization of concrete IEN, such as 10–14 years old male immigrants from a middle-income background. An important reminder is therefore to interpret strata information from the viewpoint of discriminatory accuracy, i.e., the capacity of an IEN to classify with accuracy cases and non-cases within the population. The average immigrant teenage boy is obviously not likely to have behavioural problems associated with an ADHD diagnosis. In the same way, a diagnosis may occur in any strata of the population. However, some subjects are at much greater risk, and the intersectional grouping in this study provides a detailed picture of this uneven distribution.

Since observability is a crucial vector in the terminology of Hacking [65], Brossard [42] has proposed that behaviours associated with mental disorders are primarily *role* disturbances. A diagnosis therefore requires identification of how social roles are disturbed and such an observation is contingent on social networks in the specific context. This might explain why our study illustrates a thorough heterogeneity between different strata. Furthermore, prevalence discrepancies between natives and immigrants within the low-income-group may depend on differences between countries of origin, regarding attitudes and access to psychiatric treatment. In some countries, psychiatric care is not utilized to the same extent, and parents from

other regions of the world could be expected to have differing expectations of the health care system. It is possible that psychiatric diagnoses such as ADHD may be considered more stigmatising in some cultures [66], and this could be particularly evident among people with less resources, i.e., low income. In addition, there could be quite some cultural variation in what is considered acceptable or standard childhood behaviour [67]. However, our study uses a very rough classification of country of birth, grouping all immigrants together, even though this is a highly heterogenous category. It is a limitation of this study that it was not possible to analyse how coming from a war afflicted country, trauma, education level, cultural differences, and so on, relates to our results. In future studies, we propose an analysis using a more granulated classification of country of birth. Since the study illustrated a strong overall link between ADHD and socioeconomic/demographic variables, future research could also benefit from expanding the analysis by incorporating other variables in the construction of strata, i.e., geographic area and/or specific care facility. The impact of the clinician can for example be a key factor behind varying prevalence. As commented in the introduction, the IEN-AIHDA framework is more flexible than an analysis formally based on intersectional theory only. An important limitation with the application of IEN-AIHDA in this study is that we were unable to choose the variables of interest a priori, since the information in the registers was limited. Future studies can contribute to the IEAN-AIHDA approach by adding new variables hypothesized to be relevant in the understanding of ADHD risk distribution.

So how should the massive heterogeneity in this study be interpreted in relation to other studies? Previous research has shown that lower family income is associated with stressful events [68], parenting style [69, 70], and family conflict [71]. Regarding ADHD, population-based studies have confirmed that there is a link between an increased risk of receiving a diagnosis and social problems such as childhood adversities and low family income [72, 73]. Such relationships between psychiatric disorders and social factors can be explained by either (i) a social drift hypothesis, i.e., that biologically caused illness increases the risk of drifting into a lower social class and decreases the chances for upward social mobility, *or* (ii) a social causation hypothesis, i.e., that social disadvantage increases the risk of developing psychiatric disorders [74]. When it comes to these socioeconomic disparities in ADHD, there is a risk that the social drift hypothesis will continue to appeal, since parent traits then can explain both social disadvantage and the child's symptoms, without compromising the narrative of a neurological entity, characterized by 'neural abnormalities' and a strong genetic association [75]. The biological discourse associated with behaviors labeled as ADHD, risks portraying social conditions as a dependent variable rather than a cause of illness, i.e., that individuals to a greater extend end up in socioeconomically unfavorable conditions due to neuropsychiatric characteristics, rather than vice versa. However, our large study, utilizing stratified analysis showed that in the middle and high-income groups, immigrant boys have a much higher risk of ADHD than native boys. These findings support the notion that psychosocial factors should be more highlighted since it is unlikely that children to a greater extent have an immigrant background as a result of neuropsychiatric characteristics (cf. [76]). This opens for a more thorough discussion on how social factors needs to be recognized in the understanding of behaviours and symptoms associated with receiving an ADHD diagnosis [77, 78]. Accordingly, Miller and coworkers [78] have pointed out that if socioeconomic disadvantage causes disorders such as ADHD, explanations and interventions should not only consider individual characteristics. The tendency to address social problems by seeking to detect undiagnosed biological dysfunction among socially disadvantaged youth, as noted in the introduction, is therefor problematic.

While researchers have proposed to move beyond the dichotomy of genes versus environment, and instead develop models based on interplay and epigenetics [79], the aetiologic narrative of 'ADHD symptoms' arguably continues to be unbalanced, explaining a wide range of

heterogeneous behaviors as a biological disease entity [6, 20, 80]. Te Meerman and coworkers [80], for example, found that academic textbooks were rather biased, overstating the results of twin studies, and not addressing the implications of disappointing molecular research. Such an emphasis on genetical coherence through twin studies tends to disguise the fact that environmental factors can be of great importance even for traits with a very high heritability (i.e., human height, which in Europe increased by 11 centimeters in just 100 years [81]). This is not to say that the different strata in this study provide a more coherent explanation on why individuals become diagnosed. Rather, it is important to acknowledge the complexity of diagnoses entirely based on behavioral problems, and to highlight that some psychiatric categories should be approached with care, as blunt clinical tools. In clinical practise, there is a risk that ADHD comes to serve as an umbrella term for people that suffer and display a wide variety of maladaptive symptoms: A trend that might be accentuated due to a reorientation from attention and hyperactivity, towards a broader clinical discourse on 'executive difficulties' or 'self-regulation'. In line with how medicalization processes operate, this leads to an all too wide range of behavioral difficulties in everyday life residing within the same explanatory category.

It is highly important to consider to which extent ADHD is utilized as an explanation rather than a heterogenous description [20]. In an understanding of the ontology of ADHD, the core question is what it actually means *to have ADHD*. Is it to have been diagnosed? Is it to be positioned at the outskirts of a bell-shaped curve for a certain trait? Or is it to have difficulties in everyday life, i.e., stated as 'functional impairment'? And perhaps most importantly, what are the relationships between these parameters? High-functioning adults can display tangible features of chaos, hyperactivity, inattentiveness, and impulsivity in their everyday life, without experiencing much suffering. People who have such traits to a lesser extent (but who also have childhood trauma, financial stress, low status in society, low self-esteem) arguably have a higher risk of fulfilling criteria D–problems with academic, social, and/or occupational functioning–and therefore being diagnosed with ADHD. In the end, a diagnosis is not a result of how much 'ADHD traits' a person has, but how well his or her life is working out. This functioning over time, i.e., adaptation to life, is apparently crucially intertwined with social variables. Such an awareness does not dismiss the fact that ADHD, as well as every other psychiatric category, contains a certain biological predisposition. But when it comes to understanding the increasing presence of ADHD in clinical discourse, in academic research, and in media, it is vital to distinguish between people's behavioral characteristics on the one hand, and why something becomes a disability, on the other. Even if a trait such as activity level might have a substantial genetic basis, this is not the same thing as having a disease [80]. The genetic findings from twin studies may simply display that children at risk of receiving an ADHD diagnosis more often have certain temperament factors. However, this should not be conceptualized as an explanation, since the disorder is ultimately contingent on an environment that is not sufficiently adapted to the developmental needs of the specific individual. Such an interpretation is in line with the findings of this study, where context–intersectional ecological niche–had a large impact on prevalence.

There is an abundance of research that has investigated correlates between subjects diagnosed with ADHD and potential biological abnormalities. Studies have reported an association with inflammatory processes [82], hypothesized a connection to foods and inhalants [83], or highlighted ADHD as part of a broader biological dysfunction in stress-relevant mechanisms [84]. As a nuance to this research paradigm, it is an important reminder that a person's psychiatric symptoms seldom can be categorized into those for which we can find a biological cause and those for which we can not. This is why the term *idiopathic* is not even included in the DSM. When trying to imitate somatic care by utilizing assessments and explanatory narratives centered around the body in order to find a biomedical explanation behind behavioral

symptoms, it is not only a misunderstanding of the very nature of mental disorders, but also a way of shifting focus and resources from other modes of understanding. In a time when the number of ADHD diagnoses are hitting epidemic proportions, parallel to growing economic gaps, it is important to be sociologically observant of which people that are receiving these diagnoses, and what kind of distress is being subject to 'psychiatrization' [85, 86].

## Conclusion

Our study analyses a large record linkage and nationwide database, which minimises the risk of selection bias that can be prominent in smaller studies. In summary, using IE-AIHDA we mapped prevalence of ADHD across numerous strata defined by demographics and socioeconomic categories. As far as we know, our study is pioneering in providing such information. We found a large heterogeneity in the risk for an ADHD diagnosis across those IE strata, and there is a need for further studies to understand the reason for these large differences in prevalence. For this purpose, the use of intersectional contexts and thinking of these contexts as 'ecological niches' provides a promising approach for illuminating the diagnostic trend as a process that is historically, sociologically, and culturally situated. The large differences between different social groups raises important questions. It is our hope that future studies aim to focus on identifying factors that increases the risk of labelling behavioural and social issues as ADHD. Research could also benefit from addressing contextual interpretation of symptoms and diagnosis, contradictions between different paradigms, and practical applications of the above insights.

## Supporting information

**S1 Table. Number of individuals (N) and prevalence or Absolute Risk (AR) of Attention Deficit Hyperactivity Disorder (ADHD) in the 96 intercategorical strata as well as prevalence ratios (PR).** Ranked.
(DOCX)

**S2 Table. Number of individuals (N) and prevalence or Absolute Risk (AR) of Attention Deficit Hyperactivity Disorder (ADHD) in the 96 intercategorical strata as well as prevalence ratios (PR).** Not ranked.
(XLS)

**S1 Dataset. Minimal data set.**
(DTA)

**S1 File. Syntax.**
(DO)

## Author Contributions

**Conceptualization:** Christoffer Hornborg, Juan Merlo.

**Data curation:** Raquel Pérez Vicente, Juan Merlo.

**Formal analysis:** Rebecca Axrud.

**Methodology:** Raquel Pérez Vicente, Juan Merlo.

**Project administration:** Christoffer Hornborg.

**Supervision:** Juan Merlo.

**Writing – original draft:** Christoffer Hornborg, Rebecca Axrud.

**Writing – review & editing:** Christoffer Hornborg, Rebecca Axrud, Juan Merlo.

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
