## [Decision Letter · Decision Letter 0]

13 Sep 2023

PONE-D-23-22871Socioeconomic disparities in attention deficit hyperactivity disorder (ADHD) in Sweden: An intersectional ecological niches analysis of individual heterogeneity and discriminatory accuracy (IEN-AIHDA)PLOS ONE

Dear Dr. Hornborg,

Thank you for submitting your manuscript to PLOS ONE. After careful consideration, we feel that it has merit but does not fully meet PLOS ONE’s publication criteria as it currently stands. Therefore, we invite you to submit a revised version of the manuscript that addresses the points raised during the review process.

 Kindly review the remarks and feedback provided by the reviewers attachedPlease address the feedback provided by the reviewers that recommend Major concerns firstOverall, i enjoyed the context of the study and the research question, and would like to see a revised version ==============================

We look forward to receiving your revised manuscript.

Kind regards,

Lakshit Jain, MD

Academic Editor

PLOS ONE

Journal Requirements:

Reviewers' comments:

Reviewer's Responses to Questions

**Comments to the Author**

1. Is the manuscript technically sound, and do the data support the conclusions?

Reviewer #1: Yes

Reviewer #2: Yes

Reviewer #3: Yes

Reviewer #4: Yes

Reviewer #5: Partly

Reviewer #6: Yes

Reviewer #7: Yes

2. Has the statistical analysis been performed appropriately and rigorously? 

Reviewer #1: I Don't Know

Reviewer #2: Yes

Reviewer #3: Yes

Reviewer #4: Yes

Reviewer #5: Yes

Reviewer #6: Yes

Reviewer #7: Yes

3. Have the authors made all data underlying the findings in their manuscript fully available?

Reviewer #1: Yes

Reviewer #2: Yes

Reviewer #3: Yes

Reviewer #4: Yes

Reviewer #5: Yes

Reviewer #6: Yes

Reviewer #7: Yes

4. Is the manuscript presented in an intelligible fashion and written in standard English?

Reviewer #1: Yes

Reviewer #2: Yes

Reviewer #3: Yes

Reviewer #4: Yes

Reviewer #5: Yes

Reviewer #6: Yes

Reviewer #7: Yes

5. Review Comments to the Author

Reviewer #1: 1. The study includes whole Sweden population as a sample and that is the biggest strength of this study.

2. The core question in current world still remains, what is the reason for increased diagnosis of ADHD in recent years ? Whether its really more prevalent now compared to previous decades or just were undiagnosed so far? Author has made it clear that aim of this study is different but it can't be separated form the core issue and so this study should have comment on this question too.

Reviewer #2: Very important study on a topic that highly controversial on both sides of the Atlantic! It's a luxury to have access to such huge national database as it can inform a growing trend without having to worry about sampling biases.

The background well written; however, I wonder if there's a way to make it somewhat simpler to follow. It does go into more details which could be presented even more effectively during the discussion part.

I would also comment on the fact that demographic data often look at difference from one perspective- the patient's perspective. It is equally important to consider the impact of the clinician who is making the diagnosis of ADHD. As a clinician practicing in both community and university setting, I come across many individuals with complaints of "ADHD" that have had the diagnosis made by primary care providers (or pediatricians), who might have "met the criteria" if seen by specialists. I don't highlight this point to discuss who is better placed to make this diagnosis, but only to review how clinician access (High/ low social economical status, immigrant/ native populations) could also be a major player behind varying prevalence. If that does seem to be the case, perhaps training could also help bridge this gap. If this information is not readily available, perhaps it could be highlighted as a limitation.

Another factor to consider will be grouping "immigrant" population in one cohort. Immigrant population can vary highly from the standpoint of cultural differences in seeking mental health treatment, level of tolerance in managing these symptoms or to approach them with discipline.

The overriding principle of how the diagnosis of ADHD essentially describes symptoms and groups them to "create" a medical condition, is problematic in my opinion. Looking through the same lens, diagnosis of Major Depressive Disorder might not differ much. For example the same sentence that describes ADHD in the introduction section of the study as “I have concentration difficulties because of my ADHD” is as tautological as saying “I have concentration difficulties because of my concentration difficulties,” could be applied to depression as "I have depression because of my Major Depression" as saying, "I have depression because of my depression." I only highlight this point as such statements may divert readers away from reviewing the study and understanding what it shows.

Reviewer #3: Thank you for the opprotunity to read and review this study. The study presents a new approach called IEN to better understand the distribution of ADHD. The approach involves a detailed analysis that goes beyond the national prevalence rate and reveals a significant variation in ADHD prevalence across demographic and socioeconomic categories. By adopting an intersectional perspective, the study shows that ADHD prevalence is influenced by various factors such as age, gender, immigration status, and socioeconomic standing. The study also emphasizes the importance of cautious interpretation of strata information to avoid perpetuating stigmas.

Despite the valuable insights gained from the stratified analysis, the study acknowledges the potential for stigmatization due to the detailed information.

In addition, the study challenges previous research on the link between family income and ADHD and requires further exploration of potential explanations. Overall, the study's novel methodology, stratified analysis, intersectional perspective, and responsible data interpretation are commendable. However, addressing contextual interpretation, stigmatization concerns, cultural factors, practical applications of theory, and contradictions with existing research would further enhance the study's insights into the complex ADHD distribution landscape. (This was a suggestuion for future work.)

Reviewer #4: The paper presents a comprehensive analysis of ADHD prevalence in Sweden across different demographic and socioeconomic groups. Through a large-scale examination of over six million individuals, the authors unveil significant variations in ADHD risk based on age, gender, income, and country of birth. The paper had two primary aims: understanding the demographic and socioeconomic distribution of ADHD risk in Sweden, and contributing to the ongoing discourse on medicalization, particularly in relation to behavioral and social issues. I have the following comments/questions:

• The results provide valuable epidemiological information about the individuals diagnosed with ADHD. The authors do a great job of achieving aim 1

• Although I agree with sentiment expressed by the authors about considering sociodemographic factors and their effects of ADHD diagnosis, their data does not do them any favors from an explanatory standpoint. The wide paradoxical outcomes reported across different strata do not provide a coherent explanation of how these sociodemographic factors affect the diagnosis. However instead of acknowledging this complexity and accepting that based on only there results they cannot conclusively confirm or refute the claim that increasing prevalence of ADHD can be explained away by sociodemographic factors, it appears that they double down and try and prove a point their data does not support. Thus the discussion section would benefit from such an acknowledgment

Reviewer #5: This is an interesting study that compared the risk of suffering from ADHD across 96 strata in the Swedish population.

Strengths of the study:

-The methodology of the study is strong. Being a population level study that has access to a large database (that covers 99% of inpatient and 80% of outpatient diagnoses) makes the specific results extremely well generalizable for a western country.

-Statistical analysis was also well described and is a valuable addition to the literature.

A few concerns:

-The introduction section would benefit from a rewrite and significant editing. There were multiple long run-on sentences. The same concepts were explained multiple times in different subsections of the introduction (like the discussion about theories from Ian Hacking, these are explained at length in both the 'ecological niches as a conceptual framework' and 'intersectional analysis' sub-sections). All of this could be condensed into one introduction section.

A couple of examples of sentences that need editing:

1. page 2, first paragraph of introduction: "Without questioning the intentions of the proposal- people in these vulnerable areas do have a reduced tendency to seek this type of care - there is an all the more important observation to make." Could be broken down into easier and understandable language.

2. page 5, second paragraph: "Something that started out as a label for hyperactive children, ......"

-Alternatively, I would consider moving a lot of the introduction section into the discussion. Much of the concepts explained in the introduction belong much better in the discussion rather than setting up the study. This also fits in with the stated aims of the authors which include 1. the actual study looking at the distribution of ADHD and 2. a critical discussion on diagnosis and medicalization.

-In the discussion section, the authors highlight the differences between strata and attempt to explain these differences well. The roles of social context and understanding the ontology of ADHD are explained well and contribute to the discussion around this issue.

-However, the authors fail to offer a counter perspective to their arguments. As the authors point out in the introduction section, there have been investigations into function, structural and neurotransmitter alterations in the brains of those with ADHD. While the data may not be comprehensive and overwhelming indicative of one cause, there is enough underlying biological predisposition that cannot be dismissed entirely. While the social content that the authors describe can be potentially responsible for the increase in ADHD prevalence (the so called "epidemic"), this does not dismiss ADHD as an actual neurobiological condition. Rather it just helps us look at the parameters for diagnosis more critically. The clam the authors make around there being no 'natural' prevalence of ADHD that can be found is concerning. While our current tools/parameters for diagnosis are not ideal, this just seems to indicate that decades of scientific research into ADHD is pointless. This perspective is contradicted by clinical wisdom and also by the effectiveness of stimulant medication in treating ADHD symptoms (in the absence of other comorbidities). The authors do present a valuable perspective. However, this should be tempered and cognizant of other literature. The discussion section should be better balanced and appears rather skewed towards one side of this issue.

Overall, a good study, could be a much better manuscript with some work.

Reviewer #6: Please see attached world document for manuscript review edits and comments. There are no ethical concerns for the article. Authors conclude that ADHD's association to sociological component. I congratulate authors for same.

Reviewer #7: Your paper presents some intriguing ideas, but the explanation of the intersectional ecological niche framework requires further clarification. Additionally, the summary could benefit from some concise editing. While discussing your work, individuals acknowledged its strengths, yet they expressed uncertainty regarding its limitations. Providing a more detailed explanation of these limitations is crucial.

6. PLOS authors have the option to publish the peer review history of their article (what does this mean?). If published, this will include your full peer review and any attached files.

Reviewer #1: **Yes: **VIMAL SATODIYA

Reviewer #2: **Yes: **Ankit Parmar, MD

Reviewer #3: **Yes: **Anil Bachu

Reviewer #4: No

Reviewer #5: No

Reviewer #6: No

Reviewer #7: No

---

## [Author Response · Author response to Decision Letter 0]

28 Sep 2023

Please see rebutal letter: 'Response to Reviewers'.

---

## [Decision Letter · Decision Letter 1]

17 Oct 2023

PONE-D-23-22871R1Socioeconomic disparities in attention deficit hyperactivity disorder (ADHD) in Sweden: An intersectional ecological niches analysis of individual heterogeneity and discriminatory accuracy (IEN-AIHDA)PLOS ONE

Dear Dr. Hornborg,

Thank you for submitting your manuscript to PLOS ONE. After careful consideration, we feel that it has merit but does not fully meet PLOS ONE’s publication criteria as it currently stands. Therefore, we invite you to submit a revised version of the manuscript that addresses the points raised during the review process.

ACADEMIC EDITOR: The authors have made several changes to the article and have adequately addressed the concerns raised by reviewers that evaluated the previous draft. This is reflected in 3 reviewers recommending that the article be accepted in current form.However, a new reviewer (reviewer 8) has raised additional concerns and has recommended that the article undergo major revisions.Kindly address the concerns raised by reviewer 8 and revise the article as you see fit.==============================

We look forward to receiving your revised manuscript.

Kind regards,

Lakshit Jain, MD

Academic Editor

PLOS ONE

Reviewers' comments:

Reviewer's Responses to Questions

**Comments to the Author**

1. If the authors have adequately addressed your comments raised in a previous round of review and you feel that this manuscript is now acceptable for publication, you may indicate that here to bypass the “Comments to the Author” section, enter your conflict of interest statement in the “Confidential to Editor” section, and submit your "Accept" recommendation.

Reviewer #1: All comments have been addressed

Reviewer #5: All comments have been addressed

Reviewer #7: All comments have been addressed

Reviewer #8: (No Response)

2. Is the manuscript technically sound, and do the data support the conclusions?

Reviewer #1: Yes

Reviewer #5: Yes

Reviewer #7: Yes

Reviewer #8: Partly

3. Has the statistical analysis been performed appropriately and rigorously? 

Reviewer #1: I Don't Know

Reviewer #5: Yes

Reviewer #7: Yes

Reviewer #8: Yes

4. Have the authors made all data underlying the findings in their manuscript fully available?

Reviewer #1: Yes

Reviewer #5: Yes

Reviewer #7: Yes

Reviewer #8: Yes

5. Is the manuscript presented in an intelligible fashion and written in standard English?

Reviewer #1: Yes

Reviewer #5: Yes

Reviewer #7: Yes

Reviewer #8: Yes

6. Review Comments to the Author

Reviewer #1: 1. Author has revised the manuscript with considering each reviewer comments and that has made the article better if not the best.

Reviewer #5: I appreciate the authors thoughtful response to my review.

The authors have sufficiently addressed my concerns and have introduced a more nuanced and balanced perspective.

The manuscript is much stronger as a result of the changes. I especially enjoyed reading the new discussion section. One of key points introduced in the revision is that interventions focused on ADHD should not just be limited to individual characteristics and rather focus on social conditions along with it. This is the highlight for this study.

Good job!

Reviewer #7: The authors made appropriate changes in this revision. The research presents very good insight and better understanding about the ADHD and the new methods.

Reviewer #8: 1. I would like to thank you for the opportunity to review this article. The authors have done a wonderful job gathering the data and the sample size is excellent. There are clear graphs that are self explanatory.

2. The articles shows the difference in prevalence rates in different gender, age, country of birth and income groups in Sweden. The low income group in immigrant population has the lowest prevalence whereas middle and high income group immigrant (boys) have much higher prevalence. The native Swedes Low income group have higher prevalence compared to high income group natives. The high and middle income immigrant group showed a totally different trend compared to low income immigrant group which cannot be explained. There is definitely a role of social disadvantage as stress and trauma definitely contribute to increase in prevalence of ADHD diagnosis. But that does not explain why the trend reversed. The author has tried to explain it by saying that it probably depends on people’s attitudes and access to psychiatric treatment. However the article does not explain why the trend reverses with increase of income in immigrant population. Perhaps it will be a good idea to understand where the immigrant population is coming from. Refugees from war afflicted countries with totally different cultures will have chronic stress, trauma and difficulty acculturing to the new country. Also children usually express frustration (due to difficulties with new language and culture) by showing agitation which can be confused as impulsivity and aggression. Perhaps getting information on where the immigrants came from, educational level, perception of mental health in immigrants, cultural differences from countries of origin, trauma and substance use in population will be a valuable information to obtain

SOLUTION: Perhaps the lack of above information can be included in the limitation of study

3. The article explains beautifully that incidence has increased but fails to address why is it so. The current research also takes inflammation into account for explaining increased prevalence. Perhaps the modern diet and modern day stresses in the childhood and prenatal period has something to do with it

SOLUTION: Perhaps the authors can also mention these findings in research paper. I am attaching a few reference articles for author.

References

a. Pelsser LM, Buitelaar JK, Savelkoul HF. ADHD as a (non) allergic hypersensitivity disorder: a hypothesis. Pediatr Allergy Immunol. 2009 Mar;20(2):107-12. doi: 10.1111/j.1399-3038.2008.00749.x. Epub 2008 Apr 24. PMID: 18444966.

b. Anand D, Colpo GD, Zeni G, Zeni CP and Teixeira AL (2017) Attention-Deficit/Hyperactivity Disorder And Inflammation: What Does Current Knowledge Tell Us? A Systematic Review. Front. Psychiatry 8:228. doi: 10.3389/fpsyt.2017.00228

c. Chang JP, etal Cortisol, inflammatory biomarkers and neurotrophins in children and adolescents with attention deficit hyperactivity disorder (ADHD) in Taiwan. Brain Behav Immun. 2020 Aug;88:105-113. doi: 10.1016/j.bbi.2020.05.017. Epub 2020 May 8. PMID: 32418647.

4. The current research also hints toward the role of genetics in ADHD. There are also neuroimaging studies describing the differences in brain structure of ADHD vs non ADHD children. So the author blaming it on medicalization of ADHD and considering it as behavioral and social problem is perhaps too simplistic and narrow of an explanation. ADHD also has neurobiological underpinnings to it.

SOLUTION: Please add it to conclusion section. There is a need for further understanding the genetic and neurological factors in addition to social correlates

References

a. Yadav SK et al. Genetic variations influence brain changes in patients with attention-deficit hyperactivity disorder. Transl Psychiatry. 2021 Jun 5;11(1):349. doi: 10.1038/s41398-021-01473-w. PMID: 34091591; PMCID: PMC8179928.

b. Stevens HE, Scuderi S, Collica SC, Tomasi S, Horvath TL, Vaccarino FM. Neonatal loss of FGFR2 in astroglial cells affects locomotion, sociability, working memory, and glia-neuron interactions in mice. Transl Psychiatry. 2023 Mar 11;13(1):89. doi: 10.1038/s41398-023-02372-y. PMID: 36906620; PMCID: PMC10008554.

PLEASE CONSIDER ADDRESSING THESE ISSUES.

7. PLOS authors have the option to publish the peer review history of their article (what does this mean?). If published, this will include your full peer review and any attached files.

Reviewer #1: No

Reviewer #5: No

Reviewer #7: No

Reviewer #8: **Yes: **JASLEEN KAUR MD

CONNECTICUT VALLEY HOSPITAL

MD PSYCHIATRY

FELLOWHIP: PSYCHOSOMATIC AND ADDICTION MEDICINE

---

## [Author Response · Author response to Decision Letter 1]

20 Oct 2023

Please see rebutal letter: 'Response to Reviewers'.

---

## [Decision Letter · Decision Letter 2]

8 Nov 2023

Socioeconomic disparities in attention deficit hyperactivity disorder (ADHD) in Sweden: An intersectional ecological niches analysis of individual heterogeneity and discriminatory accuracy (IEN-AIHDA)

PONE-D-23-22871R2

Dear Dr. Hornborg,

We’re pleased to inform you that your manuscript has been judged scientifically suitable for publication and will be formally accepted for publication once it meets all outstanding technical requirements.

Kind regards,

Lakshit Jain, MD

Academic Editor

PLOS ONE

Additional Editor Comments (optional):

Reviewers' comments:

Reviewer's Responses to Questions

**Comments to the Author**

1. If the authors have adequately addressed your comments raised in a previous round of review and you feel that this manuscript is now acceptable for publication, you may indicate that here to bypass the “Comments to the Author” section, enter your conflict of interest statement in the “Confidential to Editor” section, and submit your "Accept" recommendation.

Reviewer #1: All comments have been addressed

Reviewer #5: All comments have been addressed

Reviewer #7: All comments have been addressed

Reviewer #8: All comments have been addressed

Reviewer #9: All comments have been addressed

Reviewer #10: All comments have been addressed

2. Is the manuscript technically sound, and do the data support the conclusions?

Reviewer #1: Yes

Reviewer #5: Yes

Reviewer #7: Yes

Reviewer #8: Yes

Reviewer #9: Yes

Reviewer #10: Yes

3. Has the statistical analysis been performed appropriately and rigorously? 

Reviewer #1: I Don't Know

Reviewer #5: Yes

Reviewer #7: Yes

Reviewer #8: Yes

Reviewer #9: Yes

Reviewer #10: Yes

4. Have the authors made all data underlying the findings in their manuscript fully available?

Reviewer #1: Yes

Reviewer #5: Yes

Reviewer #7: Yes

Reviewer #8: Yes

Reviewer #9: Yes

Reviewer #10: Yes

5. Is the manuscript presented in an intelligible fashion and written in standard English?

Reviewer #1: Yes

Reviewer #5: Yes

Reviewer #7: Yes

Reviewer #8: Yes

Reviewer #9: Yes

Reviewer #10: Yes

6. Review Comments to the Author

Reviewer #1: All comments have been addressed during 1st revision and I don't have anything else to add or comment now on this article.

Reviewer #5: Authors have addressed all concerns. Manuscript is much stronger as a result. This will make a good addition to the literature.

Reviewer #7: In my opinion, the authors or best efforts in revising the paper based on the given suggestions. From my standpoint, the paper has been improved significantly and is now in a good state to be published. I appreciate your team work.

Reviewer #8: The authors have addressed all my concerns in the revised draft. The article is beautifully written and has a good sample size.

Reviewer #9: Thank you for the opportunity to review this piece on the socioeconomic disparities in ADHD. The authors have addressed the review comments appropriately. This article has clarified all comments including the pending comments from the reviewer satisfactorily. The paper brings a unique perspective to the etiology of ADHD and would add value to the existing literature on the topic. I recommend publishing this article as it is at this time.

Reviewer #10: The research paper aims to understand the socioeconomic disparities in ADHD risk in Sweden. The researchers analyzed the risk of ADHD across varied categories of age, gender, income, and country of birth. The findings challenge the conventional view of ADHD as primarily a neurological abnormality and suggest that there exist significant sociological factors determining how some individuals are more susceptible to ADHD. The argument that ADHD is not solely a neurological abnormality is valid and well substantiated with the results from the study which showed considerable risk heterogeneity across different demographic and socioeconomic categories. In summary, the research paper offers a compelling perspective on ADHD diagnoses, shifting the focus from a purely neurological standpoint to a sociological one.

The author has demonstrated a commendable commitment to addressing the concerns outlined in the previous review of the article. The revisions and improvements made have significantly enhanced the overall quality and clarity of the work. The paper now aligns better with academic standards and guidelines, and the arguments are more robustly supported by evidence and analysis. In particular, the author has successfully clarified points of contention, strengthened the rationale for their claims, and filled gaps in the research, thereby enriching the paper's contribution to the field. Moreover, the incorporation of specific data and empirical evidence, where needed, lends additional credibility to the assertions made in the article. The revisions also reflect a greater depth of understanding of the subject matter, providing readers with a more comprehensive and nuanced perspective. Overall, the author's responsiveness to the feedback is evident in the enhanced quality of the research, making it a more valuable and reliable addition to the academic discourse.

7. PLOS authors have the option to publish the peer review history of their article (what does this mean?). If published, this will include your full peer review and any attached files.

Reviewer #1: No

Reviewer #5: No

Reviewer #7: No

Reviewer #8: **Yes: **Jasleen Kaur MD

Reviewer #9: No

Reviewer #10: **Yes: **Aditi Sharma

---

## [Editor Report · Acceptance letter]

10 Nov 2023

PONE-D-23-22871R2 

Socioeconomic disparities in attention deficit hyperactivity disorder (ADHD) in Sweden: An intersectional ecological niches analysis of individual heterogeneity and discriminatory accuracy (IEN-AIHDA) 

Dear Dr. Hornborg:

I'm pleased to inform you that your manuscript has been deemed suitable for publication in PLOS ONE. Congratulations! Your manuscript is now with our production department. 

Kind regards, 

on behalf of

Dr. Lakshit Jain 

Academic Editor

PLOS ONE